# On the Use of Self-Supervised Speech Representations in Spontaneous Speech Synthesis

*Siyang Wang\*, Gustav Eje Henter, Joakim Gustafson, Éva Székely*

Division of Speech, Music and Hearing, KTH Royal Institute of Technology, Stockholm, Sweden

`*siyangw@kth.se`

## Abstract

Self-supervised learning (SSL) speech representations learned from large amounts of diverse, mixed-quality speech data without transcriptions are gaining ground in many speech-technology applications. Prior work has shown that SSL is an effective intermediate representation in two-stage text-to-speech (TTS) for both read and spontaneous speech. However, it is still not clear which SSL and which layer from each SSL model is most suited for spontaneous TTS. We address this shortcoming by extending the scope of comparison for SSL in spontaneous TTS to 6 different SSLs and 3 layers within each SSL. Furthermore, SSL has also shown potential in predicting the mean opinion scores (MOS) of synthesized speech, but this has only been done in read-speech MOS prediction. We extend an SSL-based MOS prediction framework previously developed for scoring read speech synthesis and evaluate its performance on synthesized spontaneous speech. All experiments are conducted twice on two different spontaneous corpora in order to find generalizable trends. Overall, we present comprehensive experimental results on the use of SSL in spontaneous TTS and MOS prediction to further quantify and understand how SSL can be used in spontaneous TTS. Audios samples: `https://www.speech.kth.se/tts-demos/sp_ssl_tts`

**Index Terms**: spontaneous speech synthesis, text-to-speech, self-supervised learning, mean-opinion-score prediction

## 1. Introduction

The availability of large amounts of data and computation has radically enhanced the capabilities of modern machine-learning systems. One way that these developments can benefit ordinary applications with smaller amounts of data and computation is via "foundation models" [1], publicly available pre-trained models created using self-supervised learning (SSL) on large amounts of unlabelled data. Models that integrate representations of speech audio learned via SSL have demonstrated impressive results in areas such as speech recognition, speaker recognition, and voice conversion [2]. Recently, these methods have also been considered for use as acoustic features in two-stage text-to-speech (TTS) [3, 4], showing promising results in replacing conventional mel-spectrogram features.

However, integrating SSL-based representations into TTS is still a novel concept, and it is not clear which representations are preferred for use in TTS, why they are preferred, nor what trade-offs are involved. In addition to differences between different models, research into other applications has found that representations from different layers of the same SSL model may be preferred for different applications [2, 5, 6]. Thus far, a handful of works have considered using SSL-based represen-

tations as TTS acoustic features [1][3, 4, 7, 8], demonstrating advantages in creating TTS systems using SSL from mixed-quality audio. SSL models have also been shown to be an effective mean opinion scores (MOS) predictor with minimal modification [9, 10]. We aim to investigate both TTS and MOS prediction using SSL models, specifically the differences among SSLs and their layers, an under-explored aspect of prior studies.

Another important shortcoming of prior studies on using SSL in either TTS or MOS prediction is that they mostly only use speech read aloud as training data. This contrasts against the majority of in-the-wild human speech, which tends to be spontaneous and unscripted. Such speech involves unique verbal and nonverbal phenomena such as breathing [11, 12], disfluencies [13], and discourse markers, which are seldom included or transcribed in conventional speech corpora, making them a blind spot of contemporary TTS research [14].

In this paper, we compare representations derived from four different speech SSLs. We selected these models based on their high scores on the SUPERB benchmark [2], similar dimensionalities and frame rates, and the availability of pre-trained weights. For some models, our comparisons consider multiple model versions, for example before and after ASR finetuning.

We study the utility of these models for two tasks in text-to-speech from spontaneous speech audio:

1. As intermediate feature representations ("acoustic features") in two-stage TTS.
2. As backbone models for automatic prediction of speech quality (MOS) of synthetic speech.

We perform comprehensive experiments on two corpora previously used for spontaneous TTS. Audio examples are available online at: `https://www.speech.kth.se/tts-demos/sp_ssl_tts`

## 2. Background

Self-supervised representations learned from large amounts of untranscribed speech audio have recently found a large number of applications all across speech technology [15]. In this section, we review the use of SSL representations in TTS and in speech-quality (MOS) prediction. A particular focus of our survey (and, indeed, our entire paper) is spontaneous and conversational speech. Despite accounting for the lion's share of human speech, and being considered vital for creating more-human like and convincing TTS for, e.g., conversational systems [16, 14], spontaneous speech and its challenges represent an underexplored topic in contemporary TTS research. On the one hand,

---

[1]We use the term "acoustic features" in a broad sense to denote any intermediate features used between stages of a two-stage or multi-stage TTS system.

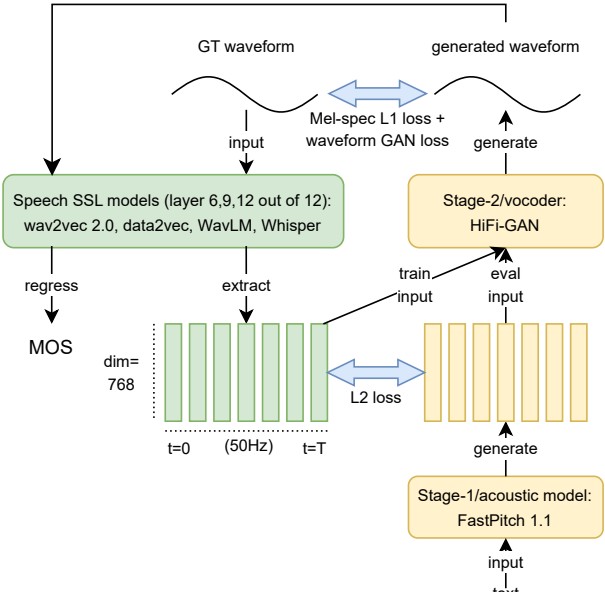

Figure 1: *Overview of this work. Two-stage TTS with SSL representations on the right and MOS prediction on the left.*

spontaneous speech exhibits increased acoustic and prosodic diversity [17], and many spontaneous-speech databases rely on found or in-the-wild recordings, which may entail reduced audio quality, competing speakers, etc. [11]. On the other hand, spontaneous speech cannot easily be partitioned into clean sentences, and the audio contains numerous phenomena that are difficult to transcribe. This includes breathing [11, 12], disfluencies such as repetitions and filled pauses (uh/um) [13], discourse markers ("like", "you know") [18]. These phenomena are often missing from the input text, but they need to be generated by the TTS system. It has been argued that these challenges of processing and synthesizing spontaneous speech can be effectively addressed by SSL [8, 19].

### 2.1. TTS Using SSL Models

The first systems using representations from modern self-supervised learning in TTS were likely WavThruVec [3] and VQTTS [4]. Both proposed to replace the acoustic representations in between the acoustic model and the vocoder with a speech feature representation from an SSL model. Compared to end-to-end TTS or traditional two-stage TTS based on mel-spectrogram features, this setup allows synthesizing high-quality audio even if the acoustic model is trained on mixed-quality audio material [3]. While VQTTS trained a custom, discrete acoustic representation (turning acoustic modelling into a classification problem), WavThruVec used a pretrained wav2vec 2.0 model [20] as its intermediate acoustic representation. There has also been work exploring the use of speech SSL models as added linguistic features instead of acoustic features [7], and as basis for discrete coarse semantic tokens in two-stage discrete-token-based TTS approaches [21].

Most relevant to the current work is the comparative study of Wang et al. [19]. They built a number of two-stage TTS systems using several publicly available speech SSL as intermediate representation, and contrasted these for synthesizing read

as well as spontaneous speech. They found that simply replacing traditionally used mel-spec with SSL representations improved both read and spontaneous TTS, but with the improvement being even more pronounced in the case of spontaneous TTS. They also found that intermediate SSL layers are better for TTS than the final layers, however they reached this comclusion by only comparing two layers in one SSL so it is limited in this respect. We focus exclusively on spontaneous speech in this work, and systematically expand the number of SSLs to 6 and the number of layers to 3 for each SSL, bringing the total number of TTS systems built to 36 (2 corpora × 6 SSLs × 3 layers) compared to only 4 in [19]

Another relevant work is MQTTS [8]. The model trains multiple discrete representations on the GigaSpeech corpus [22], which is a very large corpus of in-the-wild speech that contains a lot of spontaneous speech. Through the self-supervised learned representations, a subsequently trained TTS model is able to generate high-quality spontaneous speech, demonstrating the advantage of SSL speech representations in synthesizing spontaneous speech.

### 2.2. Quality Prediction Using SSL Models

Apart from synthesizing speech, SSL models have also been considered for predicting quality scores (specifically MOS values) of natural and synthetic speech. This was perhaps first done by [9] for predicting MOS values of different voice conversion systems. By building predictors from pre-trained SSL models and fine-tuning these end-to-end, they obtained a prediction accuracy surpassing previous state-of-the-art systems. SSLs have subsequently become a hot topic in quality prediction. The recent VoiceMOS Challenge considered predicting MOS scores of both voice conversion systems and of read-speech TTS [23], and saw a very large portion of entries that made use of SSL models. The main results saw pre-trained SSL models with fine-tuning outperform approaches that used such models without fine-tuning, in turn ahead of approaches that did not use SSL representations at all [23, 10].

Another recent challenge [24] focused on predicting speech quality in speech conferencing applications, and also saw several submissions, e.g. [25, 26], making use of SSL representations. This task does involve spontaneous speech audio, but focuses only on assessing quality of speech transmission in online conferencing and not on asessing synthesized spontaneous speech from a TTS model. Thus, none of the above works considered the use of SSL representations to predict the perceived quality of spontaneous TTS.

## 3. Method

The goal of this paper is to analyze the effect of using different SSL models in synthesizing and evaluating spontaneous speech; cf. Fig. 1. In this section we describe the SSL models studied, the data, how we build TTS systems on these data, and how we use SSL representations for subsequent MOS score prediction. Experimental results and discussion follow in Sec. 4.

### 3.1. Speech SSL Representations

Four speech SSL models were selected for our investigation. These are summarized in Table 1. All of these representations were investigated for spontaneous TTS, whereas only a subset were considered for the MOS-prediction task.

Our main reason for choosing these specific models was that they rank high on the SUPERB speech processing bench-

| SSL model (version/versions) | Training data and loss | |
| --- | --- | --- |
| | Pre-training | ASR fine-tuning |
| wav2vec 2.0 [20] (base & base-asr) | LibriSpeech 960 h (contrastive+diversity) | LibriSpeech 960 h |
| data2vec [27] (base & base-asr) | LibriSpeech 960 h (masked regression) | LibriSpeech 960 h |
| WavLM [28] (base-plus) | 94k h mixed corpora (denoising+prediction) | N/A |
| Whisper [29] (small) | N/A | 680k hours |

Table 1: *Speech SSL models tested, with info about pre-training and ASR fine-tuning corpora used.*

mark for speech SSLs [2], and have a publically available implementation and weights. Importantly, all chosen SSL models[2] have same dimensionality (765), number of layers (12 transformer layers), and frame rate (50), making them highly comparable. The main differences between the models are the data and loss function/task used for training.

For each model, we considered the representations from three different layers (6, 9, and 12 out of 12), since prior work has shown that a middle layer of SSL models contains more prosodic information [5, 19] that could benefit synthesis. For some SSL models, we also found official ASR fine-tuned versions, which we include in the experiments in addition to the self-supervised pre-training-only models. In total, we considered 18 different representations, 3 from each of 6 different SSL models.

We did not include mel-spectrogram baseline in this comparison because prior study has shown that it is much worse than SSL in two-stage spontaneous TTS [19].

### 3.2. Spontaneous Speech Corpora

We trained our TTS voices on two corpora previously used in several different studies on spontaneous TTS. The first corpus was created from the audio recordings of part 1 of the Trinity Speech-Gesture Dataset (TSGD) [30], comprising 25 monologues, each on average 10.6 minutes long, spoken by a male speaker of Hiberno English in an impromptu, colloquial style. During the recordings, the actor addresses a person seated behind the cameras, who is providing visual, but no verbal feedback. To prepare the dataset for TTS, we segmented the corpus into stretches of speech delineated by breath events following [12], and combined these segments in an overlapping fashion to form an utterance structure, with utterances no longer than 11 seconds, following [12].

The second spontaneous corpus used in this work is the ThinkComputers Corpus (TCC) [11, 14], which is a 9-hour long corpus created from the speech of one of the hosts of the ThinkComputers podcast, which is available in the public domain.[3] The podcast recordings are approximately 50 min each, and consist of two male speakers of American English discussing technology-related news and reviews. The speaking style can be described as extemporaneous, convers-

ing freely around a prepared outline, meaning that the speakers use a prepared outline, but converse freely around the planned topics. Both corpora were transcribed using ASR and subsequently corrected manually. All discourse markers, laughter, and filled pauses (uh, um) were transcribed orthographically, breath events were marked with a semi-colon, while pauses were transcribed using a comma. Other spontaneous speech phenomena such as tongue clicks were not part of the transcription.

### 3.3. TTS System

We used a similar system and training setup as WavThruVec [3], with the difference being that we omitted the multi-speaker embeddings in our system. We illustrate the system in Fig. 1.

The stage-1 model is adapted from a parallel TTS model, FastPitch [31], in which the alignment is learned automatically [32].[4] We used identical hyperparameters as in [3] to train stage-1 models, only changing the batch size to 128. We first trained on a read-speech corpus, namely LJ Speech[5], for 200 epochs, and then on each of the two spontaneous corpus for 200 epochs. This transfer-learning method has shown to be effective in allowing neural TTS to be trained on smaller spontaneous corpora [14].

For the stage-2 model or the vocoder, we used HiFi-GAN [33],[6] trained with similar hyperparameters as in [3]. We used a batch size of 160, each datum being a 0.5 second random audio excerpt. All 36 stage-2 vocoders (for the 18 SSL representations in 2 corpora) were trained for 80k steps. We used the original audio sampling rate of 22 kHz. Models of both stages were trained on 1–2 Nvidia A100 GPUs depending on batch size.

### 3.4. MOS-Prediction System

We followed [10] to build a simple wav2vec 2.0 based MOS predictor. The predictor consists of a wav2vec 2.0 base model with a mean-pooling head on top and a linear projection to a scalar MOS value. We adapted the implementation of [10][7] and followed their training procedure and hyperparameters. However, we tested different weight initializations and training-data splitting configurations for this fixed architecture, to probe how these factors affect performance of predicting MOS on spontaneous speech synthesis.

## 4. Results

This section reports and discusses our three main experimental results, namely 1) vocoder error in copy synthesis achieved by tested SSLs, 2) the design and results of subjective listening test, and 3) the performance of SSL-based MOS predictors on MOS scores collected in the subjective listening test.

### 4.1. Vocoder Error in Copy Synthesis

We conducted copy synthesis of the audios in the validation set. SSLs were extracted from ground-truth audios in the validation set and then vocoded through the correspond-

---

[2]Most models have several sizes, e.g. wav2vec2.0 base and large. We chose the base one in those cases.

[3]https://archive.org/details/podcasts_miscellaneous Creator: ThinkComputers

[4]https://github.com/NVIDIA/DeepLearningExamples/tree/master/PyTorch/SpeechSynthesis/FastPitch. This model is called "FastPitch 1.1" in this official implementation.

[5]https://keithito.com/LJ-Speech-Dataset/

[6]https://github.com/jik876/hifi-gan

[7]https://github.com/nii-yamagishilab/mos-finetune-ssl.git

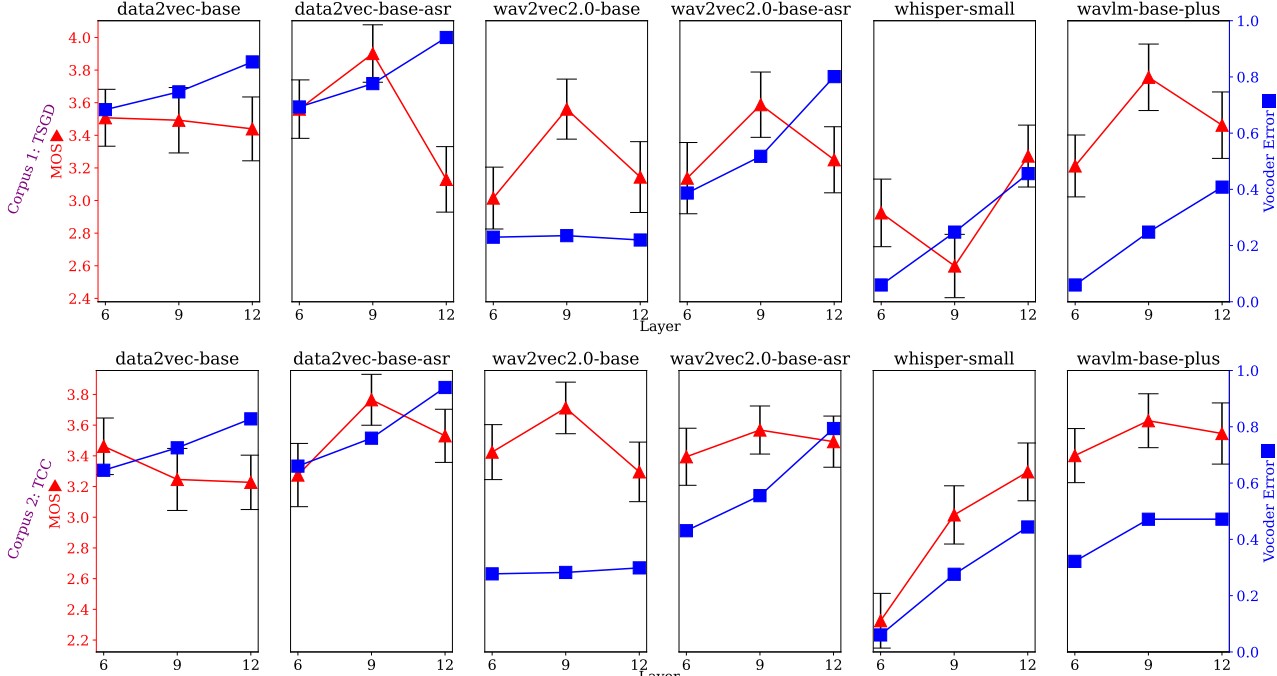

Figure 2: *Subjective MOS↑ of overall TTS pipeline and vocoder error↓ of all compared systems on the two corpora. The error bars around MOS values (red triangles) represent 95% confidence intervals.*

ing vocoders/second-stage models. Mel-spectrogram L1 error achieved by the vocoders are graphed in Fig. 2 (blue squares), showing how these errors depend on the SSL model and layer.

There are clear trends in that, the deeper the layer, the greater the error, presumably because deeper layers are further removed from the original speech waveform. In the two pairs of pre-training only and ASR fine-tuning models of data2vec and wav2vec2.0, ASR fine-tuning consistently leads to increased vocoding error over corresponding pre-trained models. However, we note that the lowest vocoding error overall is attained by representations from Whisper, which is a dedicated ASR model. In informal listening, we were impressed with the copy-synthesis performance from the Whisper-derived representations, consistent with its lowest vocoding error numbers.

Except for WavLM, the vocoding errors are very similar across the two corpora for the same model and layer. The two corpora are different in many aspects, so why are the achieved vocoding errors so close in the two corpora? This phenomenon deserves future investigation.

### 4.2. Subjective Evaluation of TTS Systems

We performed two MOS listening tests according to ITU standard P.800 [34], one for each corpus, to evaluate the full two-stage TTS pipelines built with different SSLs. Each evaluation used a pool of 20 utterances synthesised by each of the 18 different systems, for a total of 320 stimuli per corpus. For each corpus we recruited 45 self-reported native English-speaking listeners via the Prolific crowdsourcing platform. Each listener rated 51 randomly chosen stimuli from the pool balanced for SSL and layer. Participants were asked to wear headphones, and they were requested to not take the test if they had a hearing impairment. Ratings were integer values given on a scale from 1 through 5 with text labels as specified in aforementioned

ITU standard [34]. Attention checks in the form of multi-choice speech recognition tests were included to filter out unqualified test-takers. Test-takers who completed their tests too quickly to have listened to all the audio were also disqualified. This resulted 44 valid completed tests for each corpus. Participants were rewarded with an hourly wage of approximately 12 GBP with 15 or 20 minutes paid time [8], thus 3 or 4 GBP each.

Results of the two listening tests are visualized in Fig. 2 (red triangles). We observe several prominent trends. First, the 9th layer outperforms the 12th (last) and 6th (middle) layers in 4 out of 6 SSL models on both corpora. Layer 9 outperforming layer 12 is consistent with prior study on SSL layer-wise TTS performance [19], however layer 9 also outperforms layer 6 is an interesting new finding. We also find that SSLs after ASR fine-tuning obtained better ratings than corresponding SSLs prior to fine-tuning, i.e. underwent only self-supervised pre-training. For both corpora, the best performing representation is data2vec-base-asr layer 9 (TSGD: 3.90±0.18, TCC: 3.77±0.17). It is worth noting that MOS in the range of 3.90 is at the same level as current SOTA spontaneous TTS systems [8], however we do not claim that our best system is as good as a SOTA system as it is difficult to make such comparison on MOS score alone while the settings of MOS tests could be very different. We also note that consistency of the trends in SSL models and layers between the two corpora suggests that the results are likely to generalize to other spontaneous corpora.

We also see that the vocoding errors do not correlate at all with perceived TTS quality. A lower vocoding error suggests that there is more acoustic information present in the represen-

---
[8]We used 15 minutes paid time for TCC test and 20 minutes paid time for TSGD test. We slightly underestimated completion time when conducting TCC test first, thus increased expected completion which is also the paid time for TSGD test.

| | | | Zero-shot: wav2vec2.0-base-MOS [10] | wav2vec2.0-base-MOS [10] | Fine-tuned from: wav2vec2.0-base | wav2vec2.0-base-asr |
|---|---|---|---|---|---|---|
| **TSGD** | Sample | MSE↓ | 2.77 ± 0.28 | 0.35 ± 0.08 | **0.32 ± 0.08** | 0.46 ± 0.08 |
| | | LC↑ | 0.15 ± 0.08 | 0.47 ± 0.19 | **0.51 ± 0.17** | 0.30 ± 0.23 |
| | Utt. | MSE↓ | 3.60 ± 0.00 | 0.37 ± 0.02 | **0.34 ± 0.03** | **0.34 ± 0.01** |
| | | LC↑ | -0.02 ± 0.00 | **0.36 ± 0.03** | 0.32 ± 0.06 | 0.15 ± 0.09 |
| | Model | MSE↓ | 2.85 ± 0.00 | **0.40 ± 0.02** | 0.45 ± 0.01 | 0.45 ± 0.05 |
| | | LC↑ | **0.43 ± 0.00** | 0.41 ± 0.04 | 0.22 ± 0.06 | 0.17 ± 0.17 |
| | Corpus | MSE↓ | 3.07 | 0.60 | **0.44** | 0.50 |
| | | LC↑ | **0.23** | 0.12 | 0.12 | 0.04 |
| **TCC** | Sample | MSE↓ | 2.14 ± 0.15 | **0.38 ± 0.07** | 0.42 ± 0.11 | 0.50 ± 0.07 |
| | | LC↑ | 0.19 ± 0.10 | 0.50 ± 0.10 | **0.58 ± 0.13** | 0.22 ± 0.10 |
| | Utt. | MSE↓ | 2.14 ± 0.00 | 0.40 ± 0.03 | **0.38 ± 0.02** | 0.51 ± 0.07 |
| | | LC↑ | **0.28 ± 0.00** | 0.19 ± 0.09 | 0.21 ± 0.18 | -0.24 ± 0.18 |
| | Model | MSE↓ | 2.19 ± 0.00 | **0.31 ± 0.04** | 0.32 ± 0.03 | 0.40 ± 0.02 |
| | | LC↑ | 0.19 ± 0.00 | **0.53 ± 0.07** | **0.53 ± 0.03** | 0.04 ± 0.16 |
| | Corpus | MSE↓ | 3.35 | 0.64 | **0.45** | 0.47 |
| | | LC↑ | -0.08 | 0.06 | **0.10** | 0.01 |

Table 2: *Results of MOS prediction experiments. The two numbers reported for each task and predictor are mean-square-error (MSE) ↓ and linear correlation (LC) ↑. The rows are hold-out method categories. All categories except for corpus are 5-fold cross-validated, and are thus reported mean and standard deviation of the 5 runs.*

tation, however, this does not lead to better overall two-stage TTS performance as measured in subjective MOS tests. In fact, ASR fine-tuned data2vec, the best performing SSL model in two-stage TTS, consistently exhibited one of the highest vocoding errors, whereas Whisper underperformed for TTS despite having lowest vocoding errors. This suggests that there is a trade-off between the amount of acoustic information in the representation and how well can the first-stage acoustic model predict that representation from text, a phenomenon also observed in a prior study on using SSL in two-stage TTS [19]. Notably in that study, the authors found that mel-spec which achieves lowest vocoding error is the worst representation in two-stage spontaneous TTS. Another prior study reported similar results that regular TTS models have trouble findding alignemnt between mel-spec and text in spontaneous speech corpus. Our results provide further evidence to this hypothesis that there could be a trade-off between the amount of acoustic information an intermediate representation (SSL or otherwise) contains versus its achievable prediction accuracy (from text input in a TTS setting).

#### 4.3. Evaluation of Automated MOS Prediction

Using MOS data obtained in our subjective listening tests, we probed two sets of factors that may affect the generalization ability of spontaneous-speech MOS prediction with SSL: 1) the starting weights used for fine-tuning and 2) the type of unseen data (dataset split), by specifically holding out either random audio samples, or data from specific utterances (input texts), or entire TTS models, or the full corpus (i.e., training on one corpus and predicting the scores on the other). Except for at the corpus level, we performed 5-fold cross-validation for each of these experiments. In addition to fine-tuning, we also tested the zero-shot performance of the predictor from [10].

Results from the experiments on automated MOS predic-

tion are reported in Table 2. We make a number of observations from these results. First, the zero-shot model from [10] pretrained on read-speech MOS does not make meaningful predictions on this data as shown by its high MSE in all categories, however it achieves good linear correlation in some categories. Fine-tuning improved performance, with fine-tuning on top of [10] or wav2vec2.0-base performing similarly and fine-tuning on top of wav2vec2.0-base-asr performing slightly worse. Finally, although prediction MSE is low, correlations are not as strong as the numbers achieved by MOS predictors on read-speech data [10]. Several factors may contribute to this, for example that the range of MOS values in our data is quite narrow, that we have less data available than for read speech MOS, and that predicting the scores of spontaneous TTS in general may be a more challenging task than for read speech.

## 5. Conclusion and Future Work

We have compared various self-supervised speech representations in spontaneous text-to-speech and in MOS prediction on spontaneous speech synthesis, on two different corpora. We used a total of 6 different SSLs and 3 layers from each SSL, totaling 18 representations, as intermediate features in two-stage TTS. We found that representations from layer 9 of the SSL models provided better subjective TTS quality than layer 6 or layer 12 (the final layer), with the best spontaneous TTS quality achieved by layer 9 of data2vec with ASR fine-tuning. We also found that TTS subjective MOS does not correlate with the vocoding loss obtained by the SSL representation, where the high-performing TTS representations obtained some of the worst vocoding loss, and vice versa. Our results could be used as reference for SSL selection in speech synthesis tasks that utilize SSL at any capacity, and for more in-depth analysis of inter-model and layer-wise differences of SSL models in TTS

or other synthesis tasks.

We also studied the use of SSL models in predicting MOS of spontaneous speech synthesis using data obtained in our subjective listening tests. We found that zero-shot prediction from a read-speech pre-trained SSL MOS predictor performs poorly, and that fine-tuning on spontaneous MOS data is crucial for a SSL MOS predictor to have any predictive value on synthesized spontaneous speech. Compelling future work includes studying more SSLs on larger spontaneous corpora, as well as improving SSL and TTS architectures for spontaneous speech.

# 6. Acknowledgements

This work was partially supported by Digital Futures project "Advanced Adaptive Intelligent Systems", the Swedish Research Council projects "Connected" (VR-2019-05003) and "Perception of speaker stance" (VR-2020-02396), and by the Wallenberg AI, Autonomous Systems and Software Program (WASP) funded by the Knut and Alice Wallenberg Foundation. The computations were enabled by resources provided by the National Academic Infrastructure for Supercomputing in Sweden (NAISS) at Chalmers e-Commons partially funded by the Swedish Research Council through grant agreement no. 2022-06725, and by the Berzelius resource provided by the Knut and Alice Wallenberg Foundation at the National Supercomputer Centre.

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
