# OpenReview forum: "On the Use of Self-Supervised Speech Representations in Spontaneous Speech Synthesis"
_Interspeech.org/2023/Workshop/SSW — SSW12_

### Official Review · Reviewer_VQ3j · 2023-06-02
**This paper mainly investigates the use of self-supervised speech representations in spontaneous speech synthesis by experiments.**

**Rating:** 6
**Confidence:** 5

**Review:**

This paper presents a study on the use of self-supervised speech representations in spontaneous speech synthesis. The models used in this paper are existing ones. The main contribution of  this paper is to build synthesis models using different SSLs and different layers and to investigate their performance by objective/subjective evaluations. SSL-based MOS prediction was also studied.

There are some deficiencies with this paper:
1) This paper lacks the comparison between using SSL reprsentations and using conventional spectral features for acoustic models. Therefore, it is diffucult to judge the overall advantage of using SSLs for spontaneous speech synthesis.
2) The evaluation results of using SSLs for MOS prediction are not convincing enough since only one kind of synthesis model was considered in the study.

---

### Official Review · Reviewer_2b4f · 2023-06-06
**This is an interesting comparison of SSL representations for TTS. Extension to other voices/languages may be interesting.**

**Rating:** 8
**Confidence:** 4

**Review:**

Self-supervised learning (SSL) became a hot topic in several areas of speech processing. This paper aims at comparing SSL neural model layers as and intermediate representation for TTS. FastPich is used for generation the model parameters and HifiGAN as a vocoder. Some more details on the implementation would be helpful. Also data and code availability would help others to reconstruct the results and extend them to other voices/languages/speaking styles.
I have a philosophy problem with the statement: Spontaneous phenomena "are often missing from the input text, but they need to be generated by the TTS system." Is it sure, that TTS needs to generate all these phenomena? In my opinion a machine may communicate by voice but at the same time should clearly Identify itself as a machine and should not necessarily produce everything that humans do.
Also it is hard to express these phenomena as text and even harder to generate the proper representation. A sample input text-output file pair from the test setup would be useful.
Using models that were trained on huge datasets is a promising approach even with the existing risks involved.
In the current work 3 male speakers  (1 for the TSGD -Hiberno-English and 2 for  the TCC  -US English- dataset) were analyzed. That results in some limitations for the conclusions.
In the evaluation phase 20 utterances and 18 models are mentioned, so in total 20x18=360 stimuli should be applied instead of the 320 mentioned in Section 4.2.
The MOS performance in all models of Layer 6 seems to be convincing but even the highest MOS score is quite low. What is the reason for it?
Also there is no correlation between the validation loss and the MOS value. Is there any idea for the explanation?

---

### Decision · Program_Chairs · 2023-06-14

**Decision:**

Accept

**Comment:**

SSW2003 received 45 papers. The acceptance rate is 82%. We are pleased to inform you that your paper has been accepted by the SSW2023 Program Committee. Please read the reviews carefully and submit your camera-ready paper by June 28th. Most reviewers performed a detailed review. Please answer to their questions and consider their comments. Note that camera-ready papers are credited with one extra page to allow authors to consider reviewers’ suggestions. So max 7 pages in total including figures & refs.
The deadline for submitting the revised version (with full non-anonymized authors and refs!) is 28th June.